# The HACOR Score Predicts Worse in-Hospital Prognosis in Patients Hospitalized with COVID-19

**DOI:** 10.3390/jcm11123509

**Published:** 2022-06-18

**Authors:** Massimo Raffaele Mannarino, Vanessa Bianconi, Elena Cosentini, Filippo Figorilli, Costanza Natali, Giulia Cellini, Cecilia Colangelo, Francesco Giglioni, Marco Braca, Matteo Pirro

**Affiliations:** Unit of Internal Medicine, Department of Medicine and Surgery, University of Perugia, 06129 Perugia, Italy; massimo.mannarino@unipg.it (M.R.M.); e.cosentini.1991@gmail.com (E.C.); filippofigorilli@gmail.com (F.F.); costanza.natali@gmail.com (C.N.); giul.cellini@gmail.com (G.C.); cecilia.colangelo@studenti.unipg.it (C.C.); francesco.giglioni@outlook.it (F.G.); marco.braca95@gmail.com (M.B.); matteo.pirro@unipg.it (M.P.)

**Keywords:** COVID-19, non-invasive respiratory support, pneumonia, respiratory failure, HACOR score

## Abstract

Non-invasive respiratory support (NIRS) is widely used in COVID-19 patients, although high rates of NIRS failure are reported. Early detection of NIRS failure and promptly defining the need for intubation are crucial for the management of patients with acute respiratory failure (ARF). We tested the ability of the HACOR score¸ a scale based on clinical and laboratory parameters, to predict adverse outcomes in hospitalized COVID-19 patients with ARF. Four hundred patients were categorized according to high (>5) or low (≤5) HACOR scores measured at baseline and 1 h after the start of NIRS treatment. The association between a high HACOR score and either in-hospital death or the need for intubation was evaluated. NIRS was employed in 161 patients. Forty patients (10%) underwent intubation and 98 (25%) patients died. A baseline HACOR score > 5 was associated with the need for intubation or in-hospital death in the whole population (HR 4.3; *p* < 0.001), in the subgroup of patients who underwent NIRS (HR 5.2; *p* < 0.001) and in no-NIRS subgroup (HR 7.9; *p* < 0.001). In the NIRS subgroup, along with the baseline HACOR score, also 1-h HACOR score predicted NIRS failure (HR 2.6; *p* = 0.039). In conclusion, the HACOR score is a significant predictor of adverse clinical outcomes in patients with COVID-19-related ARF.

## 1. Introduction

Acute respiratory failure (ARF) is one of the most relevant clinical features of Coronavirus disease 2019 (COVID-19)-associated pneumonia. This condition often requires orotracheal intubation for invasive mechanical ventilation and the patient’s admission to an intensive care unit (ICU) [1,2]. In patients with COVID-19-related ARF without immediate indication for intubation, the use of non-invasive respiratory support (NIRS) is widely used [3] with the intent of reducing the need for intubation, thus avoiding the complications related to invasive mechanical ventilation and allowing the patients’ management in a pre-intensive hospital setting [4]. In patients with ARF, NIRS treatment was associated with a lower risk of death compared with standard oxygen therapy [5]. However, high rates of NIRS failure (i.e., need for intubation or death) are reported in COVID-19 [6]. Early detecting NIRS failure and promptly defining the need for therapeutic escalation towards intubation is crucial to ensure the most appropriate management of patients with ARF [7].

The HACOR score is a scale based on clinical and laboratory parameters, including heart rate, respiratory rate, acidosis (assessed by pH), consciousness (evaluated by Glasgow Coma Scale), and oxygenation (assessed by PaO_2_/FiO_2_ ratio). An elevated HACOR score, measured 1-h after the start of NIRS treatment, is associated with an increased risk of needing intubation and death, in non-COVID-19 patients with hypoxemic ARF [8,9,10]; thus, the HACOR score might provide possible guidance to switch from NIRS to invasive ventilation. The utility of the HACOR score (measured 1 h after NIRS initiation) in patients with COVID 19-related ARF, has been investigated by Guia et al. [11]; they observed a good performance of a HACOR score > 5 in predicting Continue Positive Airway Pressure (CPAP) failure, defined as the need for orotracheal intubation or death in COVID-19-related ARF [11]. Valencia et al. [12] found a moderate capacity of the HACOR score to predict the failure (i.e., need for mechanical ventilation and death) of high flow nasal cannula treatment in patients with COVID-19 pneumonia. Whether an early determination of the HACOR score (i.e., calculated before the initiation of NIRS treatment) might be useful for predicting unfavorable in-hospital prognosis, has not been clarified. In patients with hypoxemic ARF of various etiologies, not including COVID-19, Carrillo et al. [13] found that the baseline HACOR score was not able to predict NIRS failure [13]. However, the prognostic value of the baseline HACOR score has never been investigated in COVID-19 patients. The availability of a clinical prognostic indicator, to be used before starting NIRS, may improve the decision-making process in patients with COVID-19-related ARF, possibly avoiding an unsuccessful NIRS course.

In this prospective observational study, we investigated the ability of the HACOR score (measured at in-hospital admission and 1-h after NIRS initiation) to predict adverse clinical outcomes in hospitalized COVID-19 patients with ARF, either undergoing NIRS or not.

## 2. Materials and Methods

In this prospective observational study, COVID-19 patients with ARF referred to the Internal Medicine ward of the “Santa Maria della Misericordia Hospital” of Perugia (Italy) from December 2020 to October 2021 were consecutively enrolled. The study protocol was developed in accordance with the principles of the Helsinki Declaration and was approved by the local ethics committee. Inclusion criteria were the following: (1) age ≥ 18 years; (2) a positive result on real-time reverse-transcriptase–PCR assays testing for SARS-CoV-2 on nasal or pharyngeal swab specimens at hospital admission; (3) presence of ARF (i.e., paO_2_ < 60 mmHg on room air at sea level or the need of oxygen therapy to maintain paO_2_ ≥ 60 mmHg); (4) informed written consent. The exclusion criteria were: (1) imminent cardiorespiratory arrest; (2) immediate indication for invasive mechanical ventilation; (3) withdrawal of study consent. Data on demographic characteristics, coexisting medical conditions, current treatments, laboratory tests, and physical and instrumental examinations performed at hospital admission, were collected and registered in the medical records for each patient. Glasgow Coma Scale (GCS) and Sequential Organ Failure Assessment (SOFA) scores were calculated. Charlson Comorbidity Index (CCI) was calculated as previously reported [14]. Tests for SARS-CoV-2 on nasal or pharyngeal swab specimens were performed through RT-PCR assays (Allplex 2019-nCoV Assay, Seegene, Seoul, South Korea, or Xpert Xpress SARS-CoV-2, Cepheid, Sunnyvale, CA, USA). Arterial and venous blood samples were processed according to standard laboratory techniques in order to determine the following laboratory variables: blood gas parameters (ABL90 FLEX blood gas analyzer, Radiometer, Brønshøj, Denmark), hemoglobin, leukocyte, and platelet count (Sysmex XT-2000i, Dasit, Milano, Italy), D-dimer (BCS XP Coagulation Analyzer, Siemens, Munich, Germany), high-sensitivity cardiac troponin (hs-cTn) (UniCel DxI 800 analyzer, Beckman Coulter, Brea, CA, USA), glucose, C-reactive protein (CRP), blood urea nitrogen (BUN), creatinine, and lactate dehydrogenase (LDH) (AU5800 Clinical Chemistry System, Beckman Coulter, Brea, CA, USA). The estimated glomerular filtration rate (eGFR) was calculated through the Chronic Kidney Disease Epidemiology Collaboration (CKD-EPI) equation.

The decision to initiate NIRS (i.e., CPAP, pressure support ventilation, or high flow nasal oxygenation) treatment was made by the internist attending the patient, based on clinical judgement. Different NIRS modalities were employed: high flow nasal oxygenation (AIRVO 2, Fisher & Paykel Healthcare, Auckland, New Zealand), helmet CPAP (StarMed Ventukit—Intersurgical SpA), CPAP administered by orofacial mask (Vivo 60/65 on CPAP mode—Breas AB—Mölnlycke, Sweden), pressure support ventilation (PSV) with an oro-nasal or full-face mask (Vivo 60/65 on PSV mode—Breas AB—Mölnlycke, Sweden). The choice of NIRS modality and its titration was based on physician’s clinical judgment, device availability, and patient’s preference. The HACOR score was calculated at hospital admission and 1 h after the start of NIRS based on the following variables: heart rate, Glasgow Coma Scale, pH, paO_2_/FiO_2_, and respiratory rate 2 (Table 1). The decision to initiate invasive mechanical ventilation was made based on an overall evaluation of the clinical and blood gas analytical parameters, as indicated by the ICU physicians. Data on the clinical course and in-hospital outcomes (i.e., need for intubation, in-hospital death, and hospital discharge) were collected and registered in medical records as well.

### Statistical Analysis

The SPSS statistical package version 24.0 (SPSS Inc., Chicago, IL, USA), was used for all statistical analyses. The Shapiro test was used to verify the normality of the study variables. Categorical variables were expressed as percentages, while continuous variables were expressed as means ± standard deviation (SD) or medians (25–75 percentile). The independent samples *t*-test, Mann–Whitney U-test, and chi-squared test were used for two-group comparisons.

Hazard ratios (HRs) for the composite endpoint of either the need for intubation or in-hospital death were calculated in patients with baseline HACOR scores ≤ 5 vs. >5 through Cox proportional Hazard models by adjusting for multiple confounders (selected among those variables which were significantly different according to high vs. low HACOR score), both in the entire study population and in the NIRS and no-NIRS subgroups. The HRs for in-hospital death as a separate endpoint were also calculated. In model 1, unadjusted HRs for the composite endpoint of need for intubation/in-hospital death have been calculated in all patients and in the subgroups of patients undergoing NIRS or not. In model 2, sex, age, body mass index (BMI), and CCI have been added as covariates. Model 3 was adjusted for sex, age, BMI, CCI, current smoking, anticoagulant therapy, systolic blood pressure (SBP), leukocytes, D-dimer, hs-cTn, CRP, LDH, eGFR and SOFA score. In patients undergoing NIRS, HRs for the composite endpoint (need for intubation/in-hospital death) and for in-hospital death were calculated according to the HACOR score obtained 1 h after the start of NIRS. Adjusted event-free survival curves among patients categorized according to HACOR score ≤ 5 or >5 were elaborated. The discrimination capacity of the baseline and 1-h HACOR score was evaluated using the area under the receiver operating characteristic (ROC) curve. The discrimination capacity of the baseline and 1-h PaO_2_/FiO_2_ ratio was also evaluated.

## 3. Results

A total of 514 patients were hospitalized for COVID-19 from December 2020 to October 2021; among these, 413 met the inclusion criteria. Thirteen patients were ruled out because they met an exclusion criterion. Consequently, 400 patients were included in the study analysis. A flowchart is presented in Figure 1.

Table 2 describes the main clinical characteristics of the 400 patients, categorized according to high (>5) or low (≤5) baseline HACOR score. Patients with high HACOR score were older and had a higher BMI. A higher prevalence of smokers was observed among patients with low HACOR scores. A higher CCI and more frequent use of anticoagulants were found in patients with HACOR score > 5. Furthermore, patients with higher HACOR scores had higher SBP, diastolic blood pressure (DBP), Leukocytes, D-dimer, hs-cTn, CRP, and LDH. A higher SOFA score and reduced eGFR was observed among patients with high HACOR score.

NIRS was employed in 161 patients with rapidly increasing oxygen needs, without contraindications to NIRS and without immediate indication to invasive mechanical ventilation. Among patients undergoing NIRS, 90 underwent pressure support ventilation, with oronasal or full-face mask interface; 51 underwent CPAP, using an oronasal mask; 10 underwent helmet CPAP; 10 underwent high flow nasal oxygenation.

Forty patients (10%) underwent intubation and 98 (25%) patients died.

Table 2 shows the association between high baseline HACOR score, the need for intubation/in-hospital death and in-hospital death in the entire study population and in the NIRS and no-NIRS subgroups. A baseline HACOR score > 5 was associated with a crude 3.6-fold (*p* < 0.001) increased risk of the composite endpoint of need for intubation or in-hospital death in the entire population (Table 3; Model 1). A 3.9-fold (*p* < 0.001) and 4.3-fold (*p* < 0.001) increased risk for the composite endpoint was found in Model 2 (adjusted for sex, age, BMI, CCI) and in the fully adjusted Model 3 (Table 3). The full adjusted HRs for the composite endpoint were 5.2 (*p* < 0.001) in the subgroup of patients who underwent NIRS and 7.9 (*p* < 0.001) in the no-NIRS subgroup. High baseline HACOR score was also independently associated with an increased risk of in-hospital death, both in the total population (HR 3.5, *p* < 0.001) and in the subgroups of patients undergoing NIRS (HR 3.76, *p* = 0.019) or not (HR 9.96, *p* < 0.001) (Table 3).

In Table 4, HRs for the composite endpoint (need for intubation/in-hospital death) and in-hospital death are reported in the subgroup of patients undergoing NIRS according to the HACOR score measured 1 h after starting NIRS. Among patients undergoing NIRS, those with baseline HACOR score > 5 were 25%, whereas those with 1-h HACOR > 5 were 28% (*p* > 0.05 for difference). A high 1-h HACOR score was associated with an increased risk of NIRS failure both in the unadjusted and adjusted models (Table 4). An increased risk of in-hospital death was also observed in the patients undergoing NIRS with a 1-h HACOR score > 5.

Adjusted event-free survival curves of patients categorized according to baseline and 1-h HACOR score ≤ 5 or >5 are presented in Figure 2 and Figure 3, respectively.

The discriminatory capacity of the baseline and 1-h HACOR scores in the prediction of the composite endpoint were compared by analysing ROC curves; the area under the curve (AUC) was similar for both HACOR measurements (baseline HACOR score AUC = 0.74, *p* < 0.001; 1-h HACOR score AUC = 0.73, *p* < 0.001; *p* = 0.83 for the difference between AUCs) (Figure 4).

Baseline and 1-h HACOR scores had better discriminatory capacity in the prediction of the composite endpoint when compared, respectively, with baseline and 1-h PaO_2_/FiO_2_ ratio (baseline PaO_2_/FiO_2_ ratio AUC = 0.69, *p* = 0.049 for difference vs. baseline HACOR score; 1-h PaO_2_/FiO_2_ ratio AUC 0.68; *p* = 0.029 for difference vs. 1-h HACOR score).

## 4. Discussion

Two main findings emerge from our study: (1) the association between 1-h HACOR score and adverse clinical outcomes in hospitalized patients with COVID-19-related ARF undergoing NIRS; (2) the increased risk of either orotracheal intubation or in-hospital death in patients with baseline high HACOR score, irrespective of NIRS use.

An early assessment of NIRS efficacy is crucial to avoid potential risks of delaying intubation. The HACOR score has been proposed as a valuable tool, based on bedside clinical parameters, for the management of patients with respiratory failure [8,9]. Specifically, a HACOR score > 5 was demonstrated as a predictor of NIRS failure and increased mortality in patients with respiratory failure [8,9]. Hence, the prognostic role of the HACOR score in patients with COVID-19-related ARF warrants specific investigation.

### 4.1. One-Hour HACOR Score as Predictor of NIRS Failure in COVID-19

Only two studies investigated the usefulness of the HACOR score as a predictor of clinical outcome in patients with COVID-19-related respiratory failure undergoing NIRS. In 128 hypoxemic patients with COVID-19 undergoing CPAP, Guia et al. observed that a HACOR score > 5 was associated with an increased risk of CPAP failure [11], defined as the need for intubation or death. In a retrospective cohort of patients with COVID-19-related pneumonia undergoing respiratory support with high flow nasal cannula, Valencia et al. [12] found that the HACOR score predicted the need for intubation or death, with an optimal cut-off point of 5.5 [12]. As in previous studies, we found an association between a high 1-h HACOR score (i.e., >5) and the occurrence of either need for intubation or in-hospital death, after adjustment for confounders. Nevertheless, some features distinguish our study from previous research. First of all, we found for the first time a higher risk of in-hospital death, as an individual endpoint, in patients with COVID-19 with a high HACOR score. The prospective design of our larger study population might have contributed to explain the significant prognostic impact of the HACOR score on COVID-19 in-hospital mortality. Second, the time-dependent prognosis was evaluated in our study by Cox proportional hazard models, instead of logistic regression models [11]. Importantly, Cox proportional hazard models have more statistical power than logistic regression [15]. Third, we found that the HACOR score had a better performance in predicting the composite outcome (death/need for intubation) as compared with the simple measurement of the PaO_2_/FiO_2_ ratio. Conversely, Guia et al. found a similar performance between the HACOR score and PaO_2_/FiO_2_ ratio. The reasons for this discrepancy in findings are not easily understood. However, our observations suggest an additional prognostic value of the HACOR score, possibly by integrating measures of different conditions which contribute to the outcome of patients with COVID-19-related ARF. Finally, differently from previous studies, the prognostic role of the HACOR score was demonstrated in patients undergoing different types of NIRS (i.e., CPAP, pressure support ventilation, high flow nasal oxygenation).

### 4.2. Baseline HACOR Score as Predictor of Adverse Outcome in COVID-19

In his validation study [8] and in most subsequent works, the HACOR score was generally measured 1 or 2 h after the start of NIRS treatment. Conversely, we also assessed the HACOR score at baseline (i.e., hospital admission) in all patients with COVID-19-related ARF, regardless of NIRS use. In the NIRS subgroup, the HACOR score predicted NIRS failure (i.e., need for intubation or in-hospital death) and in-hospital mortality even if measured before the start of NIRS treatment. Our results differ from a previous study in non-COVID-19 patients by Carrillo et al. in which the baseline HACOR score did not predict subsequent NIRS failure [13]. Along with the different types of population recruited (i.e., COVID-19 vs. non-COVID ARF), additional reasons potentially explaining the different results between our study and that by Carrillo et al. might be hypothesized. One reason may be related to ARF severity. In this regard, in its validation study, the HACOR score was more accurate in predicting NIRS failure in the most severe forms of respiratory insufficiency (e.g., acute respiratory distress syndrome—ARDS, pneumonia), in which the probability of NIRS failure is higher. A higher incidence of NIRS failure in our study as compared to the study by Carrillo et al. (48% vs. 35%), could have affected the result. Another factor that may have contributed to these different results is the impact of comorbidities on the outcome. In our study, the prevalence of comorbidities was very high compared to the study by Carrillo et al., as evidenced by a median CCI of 5 vs. 1. Hypothesizing an increased predictive value of the HACOR score in patients with multiple comorbidities, we observed that a baseline HACOR score > 5 was associated with the composite endpoint with an HR of 2.4 and 4.0 in patients with CCI < 5 and CCI ≥ 5, respectively (results not shown).

In our study, the predictive value of the baseline HACOR score also emerged in the whole population, as well as in the subgroup of patients not undergoing NIRS. Indeed, in the whole population, patients with baseline HACOR score > 5 had a 4.3-fold increased risk of adverse outcome (either the need for intubation or in-hospital death) after adjustment for confounding factors, irrespective of NIRS use. Also, in the no-NIRS subgroup, a high HACOR score was associated with a higher risk of adverse outcomes. Hence, based on our results, the baseline HACOR score could be considered a useful prognostic indicator even in patients with ARF in whom treatment with NIRS is not indicated or not tolerated.

The main clinical implication of our findings is the possible usefulness of the HACOR score calculation in hospitalized COVID-19 patients with ARF as a reliable tool to predict a higher risk of worse in-hospital prognosis, even irrespective of NIRS utilization. An association between a delay in intubation and adverse outcome with increased mortality has been amply demonstrated in patients with respiratory failure [16,17]. Therefore, early identification of patients who deserve intensive treatment is critical in the management of COVID-19 patients with hypoxemic respiratory failure.

Many clinical, laboratory, or instrumental parameters have been studied as possible predictors of adverse outcomes in patients with COVID-19 [18,19]. The availability of rapid, simple, and bedside parameters, integrated in the HACOR score, may provide valuable prognostic information that may be useful to tailor the therapeutic management and escalation of the intensity of medical care in hospitalized COVID-19 patients, especially in clinical settings with limited or overloaded resources.

This study has limitations that must be mentioned. The decision about whether to intubate a patient depends on a myriad of confounding factors, including comorbidities, ability to tolerate NIRS, and ICU bed availability, many of which were not fully considered in our study. However, our results were robustly confirmed, even after adjustment for many confounding factors, including age, blood pressure levels, comorbidities, and severity of organ dysfunction. Furthermore, our study differs, in terms of the clinical setting, from the previous ones in which the HACOR score has been validated. In fact, it was conducted in an Internal Medicine ward, at the height of the pandemic, and not in an intensive care unit. Although one of the strengths of the study lies precisely in the fact that the HACOR score was applied in a pre-intensive setting, many differences in terms of staff training, and available infrastructure could have affected the results. Hence, applying the conclusions of our work to other types of patients and to different clinical settings would be considered a hasty generalization. Finally, the number of patients undergoing NIRS (*n* = 161) is relatively small, thus requiring further confirmation in larger data samples. The heterogeneous distribution of the different types of NIRS treatment is a further limitation, as each NIRS modality has been used in small groups of patients (Table 2). Consequently, the statistical power of subgroup analysis is limited and did not allow for evaluating the effect on the outcome of the different NIRS modalities.

## 5. Conclusions

Our study supports the role of the HACOR score as an independent predictor of adverse outcomes in patients with COVID-19-related respiratory failure. The HACOR score could be a useful tool to ensure timely and correct decisions about the intensity of ventilation support in patients with respiratory failure.

## Figures and Tables

**Figure 1 jcm-11-03509-f001:**
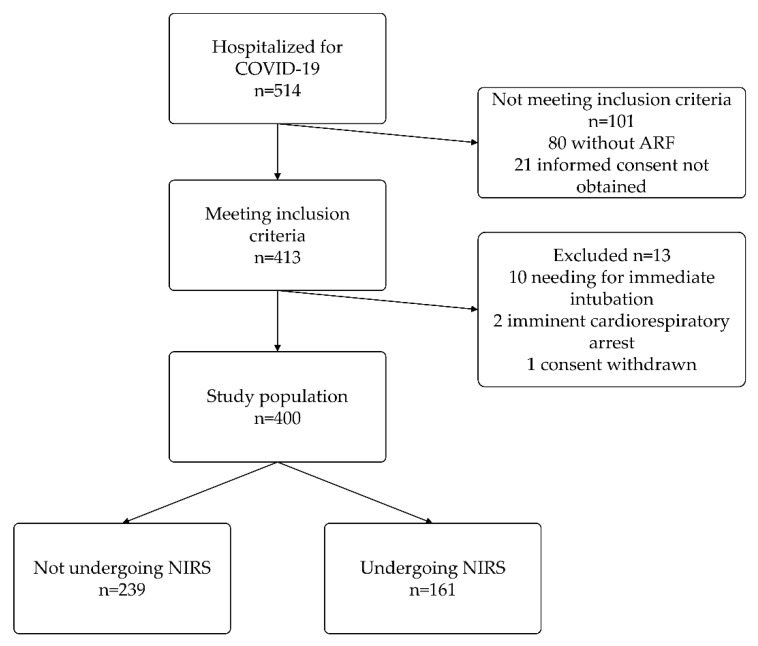
Flowchart of patients’ enrollment and NIRS use. ARF, acute respiratory failure; NIRS, non-invasive respiratory support.

**Figure 2 jcm-11-03509-f002:**
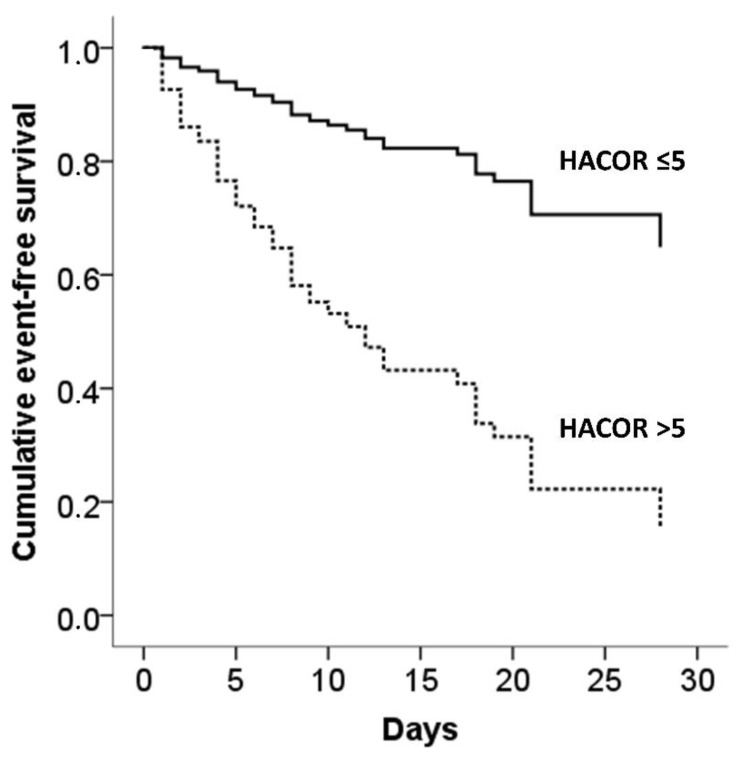
Adjusted event-free survival (need for intubation/in-hospital death) according to baseline HACOR score in the entire population.

**Figure 3 jcm-11-03509-f003:**
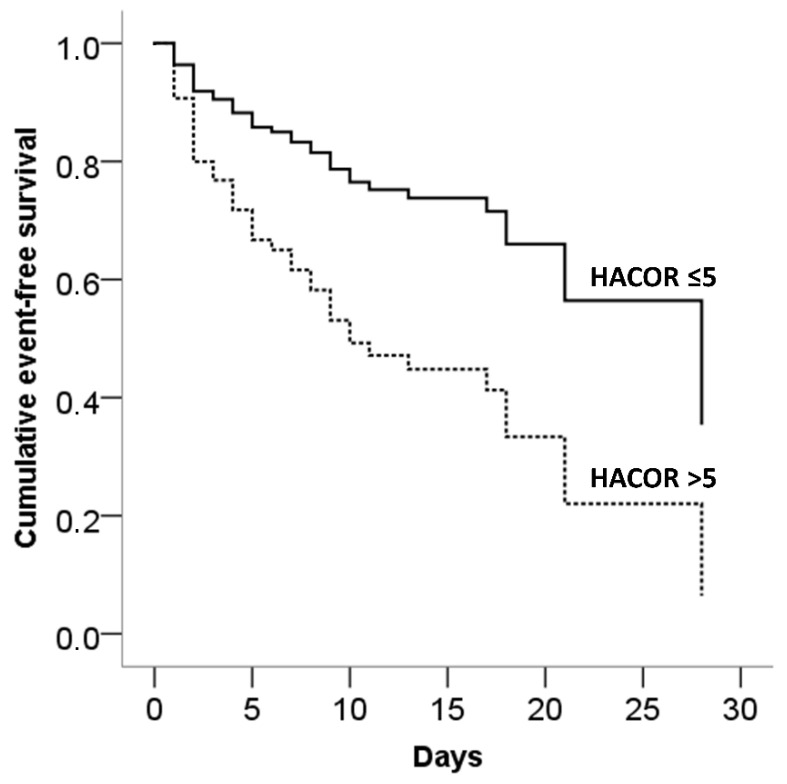
Adjusted event-free survival (need for intubation/in-hospital death) according to HACOR score calculated 1 h after starting NIRS treatment.

**Figure 4 jcm-11-03509-f004:**
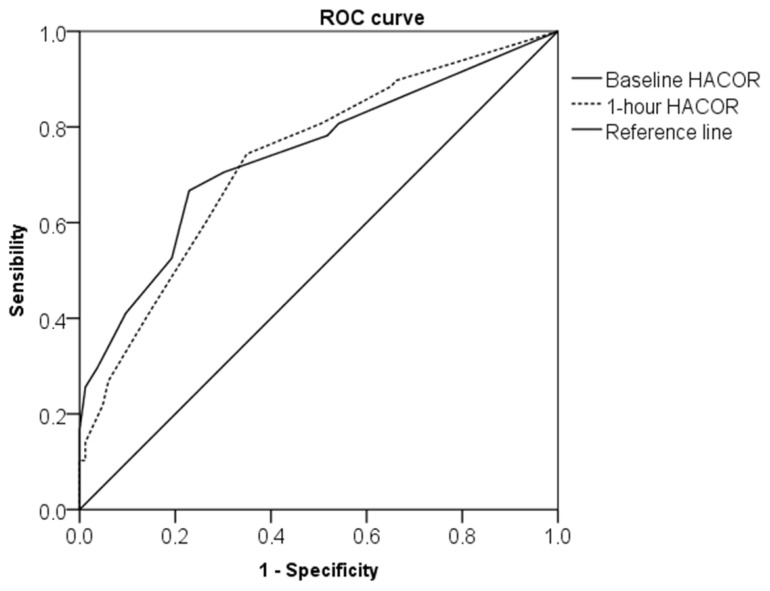
Comparison of ROC curve of baseline HACOR and 1-h HACOR scores in the discrimination of the composite endpoint (either the need for intubation or in-hospital death) in patients undergoing NIRS (baseline HACOR score AUC 0.74; *p* < 0.001 vs. 1-h HACOR score AUC 0.73, *p* < 0.001; *p* = 0.83 for difference between AUC).

**Table 1 jcm-11-03509-t001:** The HACOR score.

Variables	Values	Points
**Heart rate, beats per minute**	≤120	0
	>120	1
**pH**	≥7.35	0
	7.30–7.34	2
	7.25–7.29	3
	<7.25	4
**Glasgow coma scale**	15	0
	13–14	2
	11–12	5
	≤10	10
**PaO_2_/FiO_2_ ratio**	≥201	0
	176–200	2
	151–175	3
	126–150	4
	101–125	5
	≤100	6
**Respiratory rate, breaths per minute**	≤30	0
	31–35	1
	36–40	2
	41–45	3
	≥46	4

**Table 2 jcm-11-03509-t002:** Characteristics of the study population.

	Total (*n* = 400)	Baseline HACOR Score > 5 (*n* = 81)	Baseline HACOR Score ≤ 5 (*n* = 319)	*p*
Age, mean (SD), years	75 (14)	82 (11)	73 (15)	<0.001
Male gender, No. (%)	218 (55)	44 (53)	169 (55)	0.775
BMI, mean (SD), kg/m^2^	27 (8)	25 (4)	27 (8)	<0.001
Current smoking, No. (%)	54 (16)	7 (7)	55 (17)	0.012
Hypertension, No. (%)	251 (63)	51 (63)	193 (63)	0.965
Type 2 diabetes, No. (%)	79 (20)	16 (20)	62 (20)	0.999
Obesity, No. (%)	125 (32)	19 (22)	105 (34)	0.023
CKD, No. (%)	47 (12)	9 (10)	37 (12)	0.538
Previous CV event, No. (%)	66 (17)	15 (19)	51 (16)	0.654
Active cancer, No. (%)	33 (8)	9 (11)	22 (8)	0.349
Previous VTE, No. (%)	13 (3)	3 (4)	9 (3)	0.807
AF, No. (%)	58 (14)	12 (17)	42 (14)	0.454
COPD, No. (%)	48 (12)	10 (12)	36 (12)	0.916
CCI, median (IQR)	4 (3–6)	5 (4–7)	4 (2–6)	<0.001
Anti-hypertensives, No. (%)	260 (67)	57 (70)	203 (66)	0.464
Statins, No. (%)	69 (18)	14 (16)	56 (18)	0.680
Anticoagulants, No. (%)	127 (33)	38 (46)	90 (29)	0.009
Anti-platelets, No. (%)	100 (25)	26 (32)	71 (23)	0.123
Insulin, No. (%)	46 (12)	9 (10)	37 (12)	0.703
Oral hypoglycemic agents, No. (%)	43 (11)	9 (11)	34 (10)	0.841
SBP, mean (SD), mmHg	131 (20)	125 (23)	132 (19)	0.012
DBP, mean (SD), mmHg	76 (11)	74 (12)	77 (11)	0.072
PaO_2_/FiO_2_, median (IQR)	227 (156–276)	119 (91–155)	247 (195–284)	<0.001
Hb, median (IQR), g/dL	13.3 (11.9–14.5)	13.1 (11.4–14.4)	13.5 (12–14.5)	0.364
Leukocytes, median (IQR), ×10^3^/μL	7.5 (5.3–11.0)	9.5 (6.6–13.2)	7.1 (5–10.3)	<0.001
Platelets, median (IQR), ×10^3^/μL	208 (154–270)	212 (144–283)	202 (154–264)	0.825
D-dimer, median (IQR), ng/mL	936 (578–1788)	1389 (724–2939)	891 (571–1649)	0.003
hs-cTn, median (IQR), ng/L	14.3 (7.6–32.7)	28.3 (12.8–44)	13 (7–26.6)	<0.001
CRP, median (IQR), mg/dL	7.6 (4–13.3)	10.1 (4.6–16.9)	7.3 (3.9–12.2)	0.025
Fasting glucose, median (IQR), mg/dL	124 (106–154)	132 (110–158)	123 (106–154)	0.263
eGFR, mean (SD), mL/min	69 (25)	57 (26)	72 (24)	<0.001
LDH, median (IQR), UI/L	323 (245–431)	400 (255–474)	313 (245–419)	0.019
SOFA score, median (IQR)	3 (2–4)	5 (4–6)	2 (2–4)	<0.001
NIRS, No. (%)	161 (40)	40 (49)	121 (38)	0.068
*PSV, No (%)*	*90 (23)*	*27 (33)*	*63 (16)*	*0.083*
*CPAP, No (%)*	*51 (13)*	*9 (11)*	*42 (13)*	*0.298*
*Helmet-CPAP, No (%)*	*10 (3)*	*2 (2)*	*8 (3)*	*0.716*
*High flow nasal oxygenation, No (%)*	*10 (3)*	*1 (1)*	*9 (3)*	*0.157*

AF, atrial fibrillation; BMI, body mass index; CCI, Charlson comorbidity index; CKD, chronic kidney disease; COPD, chronic obstructive pulmonary disease; CPAP, continue positive airway pressure; CRP, C-reactive protein; CV, cardiovascular; DBP, diastolic blood pressure; eGFR, estimated glomerular filtration rate; HACOR, heart rate, acidosis, consciousness, oxygenation, and respiratory rate; Hb, hemoglobin; hs-cTn, high sensitivity cardiac troponin; IQR, interquartile range; LDH lactate dehydrogenase; NIRS, non-invasive respiratory support; PSV, pressure support ventilation; SBP, systolic blood pressure; SD, standard deviation; SOFA, Sequential Organ Failure Assessment; VTE, venous thromboembolism.

**Table 3 jcm-11-03509-t003:** Association between high baseline HACOR score, the need for intubation/in-hospital death, and in-hospital death in the entire study population and in the NIRS and no-NIRS subgroups.

	Number of Patients with Baseline HACOR ≤ 5 with/without Event	Number of Patients with Baseline HACOR > 5 with/without Event	Model 1 HR (95% CI) *p* Value	Model 2 HR (95% CI) *p* Value	Model 3 HR (95% CI) *p* Value
Need for intubation/in-hospital death					
All patients	70/249	55/26	3.62 (2.54, 5.16) *p* < 0.001	3.91 (2.62, 5.83) *p* < 0.001	4.32 (2.35, 7.92) *p* < 0.001
NIRS	46/75	32/8	2.76 (1.75, 4.34) *p* < 0.001	3.18 (1.88, 5.37) *p* < 0.001	5.17 (2.05, 13.02) *p* < 0.001
No-NIRS	24/174	23/18	4.75 (2.68, 8.43) *p* < 0.001	3.70 (1.99, 6.87) *p* < 0.001	7.87 (2.42, 25.63) *p* < 0.001
In-hospital death					
All patients	47/272	51/30	4.91 (3.30, 7.30) *p* < 0.001	4.03 (2.60, 6.22) *p* < 0.001	3.50 (1.83, 6.70) *p* < 0.001
NIRS	26/95	28/12	4.15 (2.43, 7.09) *p* < 0.001	3.70 (1.97, 6.95) *p* < 0.001	3.76 (1.24, 11.40) *p* = 0.019
No-NIRS	21/177	23/18	5.56 (3.07, 10.05) *p* < 0.001	3.77 (2.02, 7.04) *p* < 0.001	9.96 (2.86, 34.66) *p* < 0.001

Model 1: unadjusted. Model 2: adjusted for sex, age, BMI, and CCI. Model 3: adjusted for sex, age, BMI, current smoking, CCI, anticoagulant therapy, SBP, leukocytes, D-dimer, hs-cTn, CRP, LDH, eGFR and SOFA score. BMI, body mass index; CCI, Charlson comorbidity index; CI, confidence interval; CRP, C-reactive protein; eGFR, estimated glomerular filtration rate; HACOR, heart rate, acidosis, consciousness, oxygenation, and respiratory rate; hs-cTn, high sensitivity cardiac troponin; HR, hazard ratio; LDH, lactate dehydrogenase; NIRS, non-invasive respiratory support; SBP, systolic blood pressure.

**Table 4 jcm-11-03509-t004:** Association between high HACOR score measured 1 h after the start of NIRS treatment, the composite endpoint (need for intubation/in-hospital death), and in-hospital death.

	Number of Patients with 1-h HACOR ≤ 5 with/without Event	Number of Patients with 1-h HACOR > 5 with/without Event	Model 1 HR (95% CI) *p* Value	Model 2 HR (95% CI) *p* Value	Model 3 HR (95% CI) *p* Value
Need for intubation/in-hospital death	46/71	32/12	2.17 (1.38, 3.42) *p* < 0.001	2.50 (1.45, 4.30) *p* < 0.001	2.64 (1.05, 6.65) *p* = 0.039
In-hospital death	29/88	25/19	2.51 (1.46, 4.31) *p* < 0.001	2.23 (1.19, 4.19) *p* = 0.012	4.37 (1.34, 14.23) *p* = 0.014

Models as in Table 2.

## Data Availability

The data presented in this study are available on request from the corresponding author.

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
