# Peer review of "The HACOR Score Predicts Worse in-Hospital Prognosis in Patients Hospitalized with COVID-19"

_jcm, 2022, doi:10.3390/jcm11123509_

Round 1
Reviewer 1 Report
Review of manuscript no. jcm-1736866, entitled: “The HACOR score predicts worse in-hospital prognosis in patients hospitalized with COVID-19”
Reviewed manuscript has demonstrated new information about possible role of the HACOR score¸ a scale based on clinical and laboratory parameters, to predict adverse outcome in hospitalized COVID-19 patients with acute respiratory failure. The association between high HACOR score and either in hospital death or the need for intubation was evaluated. Authors demonstrated that The HACOR score could be a useful tool to ensure timely and correct decisions about the intensity of ventilation support in patients with respiratory failure.
All parts of article are composed correctly, with appropriate information in them. Material and methods, which were used in this research, were collected accordingly to the aim of the study. Statistical analysis was conducted correctly and results were demonstrated clearly and precisely. Discussion was described on the basis of appropriate earlier analyzes and own studies, and provides valuable conclusions.
Author Response
Response to Reviewer 1 comments.
Reviewer 1 comment. Review of manuscript no. jcm-1736866, entitled: “The HACOR score predicts worse in-hospital prognosis in patients hospitalized with COVID-19”
Reviewed manuscript has demonstrated new information about possible role of the HACOR score¸ a scale based on clinical and laboratory parameters, to predict adverse outcome in hospitalized COVID-19 patients with acute respiratory failure. The association between high HACOR score and either in hospital death or the need for intubation was evaluated. Authors demonstrated that The HACOR score could be a useful tool to ensure timely and correct decisions about the intensity of ventilation support in patients with respiratory failure.
All parts of article are composed correctly, with appropriate information in them. Material and methods, which were used in this research, were collected accordingly to the aim of the study. Statistical analysis was conducted correctly and results were demonstrated clearly and precisely. Discussion was described on the basis of appropriate earlier analyzes and own studies, and provides valuable conclusions.
Response: We thank the Reviewer for his/her positive comments. The manuscript has been extensively edited to improve the English language.
Changes are reported in red in the revised manuscript.
Reviewer 2 Report
This manuscript sets out to evaluate the utility of the HACOR score for predicting failure of non-invasive respiratory support prior to initiation through a single-center study.
The authors need to include the HACOR scoring rubric. Several places they list factors that enter into the score but they never actually describe the scoring.
The authors also should note the exclusion criteria and how many subjects that met the inclusion criteria had to be excluded and how many declined to consent. This information will help the reader evaluate the paper.
The methods section should also include if the data were collected prospective or retrospectively.
Given the large difference in P/F ratio between the groups with low and high HACOR scores the authors should do an analysis to see if the HACOR score is a better predictor of the combined death/intubation endpoint compared to P/F ratio. It is well established through the citations throughout this paper that patients with a P/F ratio <200 are more likely to fail non-invasive ventilation. If the HACOR score is only reflecting the P/F ratio and does not improve prediction above just looking at the P/F ratio it is not clinically useful in COVID patients prior to NIRS initiation.
The authors cite Guia et al (citation 11) as noting good performance of the HACOR score without noting that the conclusion of Guia et al manuscript is that P/F ratio is a better predictor than the 1 hour HACOR score in patients with COVID-19.
Please give a fuller description of the types of non-invasive respiratory support used at this hospital. In order for the reader to determine if the population studied is similar to their own patients they need to know what types of non-invasive respiratory support were failed. It would be helpful to see a breakdown of how subjects were distributed between the different modalities of non-invasive respiratory support and how may subjects failed more than one modality of non-invasive respiratory support. The authors mention CPAP, pressure support ventilation and HFNC but are the CPAP and BPAP delivered by nasal mask? full-face mask? helmet?
With only 161 subjects that received non-invasive respiratory support the size of the study should be included as a limitation in terms of looking at the ability of HACOR to predict failure of non-invasive respiratory support as this is the original aim of the paper as stated in the abstract.
Citation 4 does not support the sentence starting on line 33 and ending on line 36.
Citation 7 does not support the sentence starting on line 38 and ending on line 41.
Author Response
Response to Reviewer 2 comments.
This manuscript sets out to evaluate the utility of the HACOR score for predicting failure of non-invasive respiratory support prior to initiation through a single-center study.
Response: We thank the Reviewer for his/her valuable comments and constructive suggestions. We emended the manuscript in accordance with the reviewer’s requests. All changes are detailed below.
Reviewer Query: The authors need to include the HACOR scoring rubric. Several places they list factors that enter into the score but they never actually describe the scoring.
Response: The HACOR scoring has been described in a dedicated table (Table 1).
Q.: The authors also should note the exclusion criteria and how many subjects that met the inclusion criteria had to be excluded and how many declined to consent. This information will help the reader evaluate the paper.
R.: We added the requested information and we presented a flow-chart of patients enrollment in an additional figure (Figure 1).
Q.: The methods section should also include if the data were collected prospective or retrospectively.
R.: Although we stated in the introduction that this was a prospective study, this information was not reported in the methods, as the Reviewer correctly observed. Hence, we added this information also in the section "Materials and Methods" (line 67).
Q.: Given the large difference in P/F ratio between the groups with low and high HACOR scores the authors should do an analysis to see if the HACOR score is a better predictor of the combined death/intubation endpoint compared to P/F ratio. It is well established through the citations throughout this paper that patients with a P/F ratio <200 are more likely to fail non-invasive ventilation. If the HACOR score is only reflecting the P/F ratio and does not improve prediction above just looking at the P/F ratio it is not clinically useful in COVID patients prior to NIRS initiation.
We thank the Reviewer for his / her valuable suggestion. We compared the discriminatory capacity of the HACOR score and the P/F ratio with respect to the clinical endpoint; we found that both the baseline HACOR score and the 1-hour HACOR score have a better predictive capacity than the baseline and 1-hour PF, respectively. We have added this analysis in the methods (line 166-167) and results (line 231-235) and commented on it in the discussion (line 265-271).
Q.: The authors cite Guia et al (citation 11) as noting good performance of the HACOR score without noting that the conclusion of Guia et al manuscript is that P/F ratio is a better predictor than the 1 hour HACOR score in patients with COVID-19.
R.: Thanks for this valuable suggestion. The findings by Guia et al. has been commented and compared with ours in the discussion (line 265-271).
Q.: Please give a fuller description of the types of non-invasive respiratory support used at this hospital. In order for the reader to determine if the population studied is similar to their own patients they need to know what types of non-invasive respiratory support were failed. It would be helpful to see a breakdown of how subjects were distributed between the different modalities of non-invasive respiratory support and how may subjects failed more than one modality of non-invasive respiratory support. The authors mention CPAP, pressure support ventilation and HFNC but are the CPAP and BPAP delivered by nasal mask? full-face mask? helmet?
R.: As requested by the Reviewer, the different NIRS modalities have been described in more detail in the Methods (line 97-103). Furthermore, the different modalities have been reported in Table 2 and in the Results (line 157-159). CPAP and BPAP (PSV) delivered by oronasal were the most used modalities. As can be seen, there were no significant differences in terms of NIRS types between subjects with high or low HACOR scores. Furthermore, although the sample size does not allow a reliable subgroup analysis, when the type of NIRS was forced in the multivarable Cox models as a covariate, the result on outcomes was not affected.
Q.: With only 161 subjects that received non-invasive respiratory support the size of the study should be included as a limitation in terms of looking at the ability of HACOR to predict failure of non-invasive respiratory support as this is the original aim of the paper as stated in the abstract.
R.: Small sample size has been included as a limitation (line 332-335).
Q.: Citation 4 does not support the sentence starting on line 33 and ending on line 36.
The Reviewer is right, reference 4 was wrong. The sentence has been slightly changed and the reference has been replaced.
Q.: Citation 7 does not support the sentence starting on line 38 and ending on line 41.
The sentence has been slightly changed and the reference has been replaced.
The manuscript has been extesively edited to improve the English language.
Changes are reported in red in the revised manuscript.
Round 2
Reviewer 2 Report
Citation 6 does not support the sentence on lines 37 and 38, it is all about risk to healthcare workers when using NIRS on COVID-19 patients and does not have any data or citations regarding rates of NIRS failure although it does say that the rate is high in an uncited sentence.
The authors are to be commended on their consenting process. I have never seen a study with a 100% consent rate initially and a 99.8% consent rate at the end of the study, especially in a study that would need to be consented very quickly to not impede patient care with presumably a number of subjects admitted in the middle of the night in a condition where they might not be able to provide written consent themselves. Or was this study done under a consenting process other than signed consent by the subject prior to initiation of any study procedures (ie data collection)?
The added analysis comparing HACOR score with P/F ratio and the discussion comparing the results of this paper with the existing literature greatly improves the impact of this manuscript.
Citations 18-22 do not add to this paper.
Author Response
Response to Reviewer 2 Comments (Round 2)
Point 1. Citation 6 does not support the sentence on lines 37 and 38, it is all about risk to healthcare workers when using NIRS on COVID-19 patients and does not have any data or citations regarding rates of NIRS failure although it does say that the rate is high in an uncited sentence.
Response.: We thank the reviewer for the careful re-evaluation of our work.
Reference 6 has been replaced by the following:
[6] Weerakkody S, Arina P, Glenister J, Cottrell S, Boscaini-Gilroy G, Singer M, Montgomery HE. Non-invasive respiratory support in the management of acute COVID-19 pneumonia: considerations for clinical practice and priorities for research. Lancet Respir Med. 2022 Feb;10(2):199-213. doi: 10.1016/S2213-2600(21)00414-8. Epub 2021 Nov 9. Erratum in: Lancet Respir Med. 2021 Dec;9(12):e114. PMID: 34767767; PMCID: PMC8577844.
Point 2. The authors are to be commended on their consenting process. I have never seen a study with a 100% consent rate initially and a 99.8% consent rate at the end of the study, especially in a study that would need to be consented very quickly to not impede patient care with presumably a number of subjects admitted in the middle of the night in a condition where they might not be able to provide written consent themselves. Or was this study done under a consenting process other than signed consent by the subject prior to initiation of any study procedures (ie data collection)?
Response: We thank the Reviewer for the constructive comment. We recognize that the flow-chart of patients enrollment was not very clear. In fact, among the 101 patients excluded from the study for not meeting the inclusion criteria were also included those patients for whom informed consent was not obtained (the Reviewer may wish to find this specific inclusion criterion already reported in the section Methods). Thus, in order to be more clear in this regard, we added the requested information in figure 1, reporting the number of patients for each unmet inclusion criterion.
Ponit 3. The added analysis comparing HACOR score with P/F ratio and the discussion comparing the results of this paper with the existing literature greatly improves the impact of this manuscript.
Response: Thanks again to the Reviewer for the valuable suggestion.
Point 4. Citations 18-22 do not add to this paper.
Response: References [18-22] have been removed and the following one has been added:
[19] Gallo Marin B, Aghagoli G, Lavine K, Yang L, Siff EJ, Chiang SS, Salazar-Mather TP, Dumenco L, Savaria MC, Aung SN, Flanigan T, Michelow IC. Predictors of COVID-19 severity: A literature review. Rev Med Virol. 2021 Jan;31(1):1-10. doi: 10.1002/rmv.2146. Epub 2020 Jul 30.